# Transient Isolated, Idiopathic Growth Hormone Deficiency—A Self-Limiting Pediatric Disease with Male Predominance or a Diagnosis Based on Uncertain Criteria? Lesson from 20 Years’ Real-World Experience with Retesting at One Center

**DOI:** 10.3390/ijms25115739

**Published:** 2024-05-24

**Authors:** Joanna Smyczyńska, Maciej Hilczer, Urszula Smyczyńska, Andrzej Lewiński, Renata Stawerska

**Affiliations:** 1Department of Pediatrics, Diabetology, Endocrinology and Nephrology, Medical University of Lodz, 90-419 Lodz, Poland; 2Department of Endocrinology and Metabolic Diseases, Polish Mother’s Memorial Hospital–Research Institute, 93-338 Lodz, Poland; maciej.hilczer@umed.lodz.pl (M.H.); andrzej.lewinski@umed.lodz.pl (A.L.); renata.stawerska@umed.lodz.pl (R.S.); 3Department of Biostatistics and Translational Medicine, Medical University of Lodz, 90-419 Lodz, Poland; urszula.smyczynska@umed.lodz.pl; 4Department of Pediatric Endocrinology, Medical University of Lodz, 90-419 Lodz, Poland

**Keywords:** growth hormone deficiency, growth hormone therapy, near-final height, retesting, insulin-like growth factor-1

## Abstract

In the majority of children with growth hormone (GH) deficiency (GHD), normal GH secretion may occur before the attainment of final height. The aim of the study was to assess the incidence of persistent and transient GHD and the effectiveness of recombined human GH (rhGH) therapy in children with isolated, idiopathic GHD with respect to the moment of therapy withdrawal and according to different diagnostic criteria of GHD. The analysis included 260 patients (173 boys, 87 girls) with isolated, idiopathic GHD who had completed rhGH therapy and who had been reassessed for GH and IGF-1 secretion. The incidence of transient GHD with respect to different pre- and post-treatment criteria was compared together with the assessment of GH therapy effectiveness. The incidence of transient GHD, even with respect to pediatric criteria, was very high. Normal GH secretion occurred before the attainment of near-final height. Application of more restricted criteria decreased the number of children diagnosed with GHD but not the incidence of transient GHD among them. Poor response to GH therapy was observed mainly in the patients with normal IGF-1 before treatment, suggesting that their diagnosis of GHD may have been a false positive. Further efforts should be made to avoid the overdiagnosis GHD and the overtreatment of patients.

## 1. Introduction

According to current recommendations, the goals of growth hormone (GH) therapy in patients with GH deficiency (GHD) include not only the promotion of linear growth in childhood but also the correction of a wide spectrum of metabolic disorders caused by GHD and an improvement in the quality of life in all its periods. This became a premise for continuing treatment with recombined human GH (rhGH) in adulthood and for rhGH administration in the patients with adult-onset GHD. However, therapy beyond the period of growing is indicated only in the patients with severe GHD and requires the use of lower rhGH doses than in childhood. As peak bone mass and adult muscle composition is reached a few years later than the attainment of final height (FH), continuation of treatment with appropriate rhGH doses at this time—referred to as the transition period—is of particular importance if GH secretion is insufficient [1,2]. Thus, it is recommended that GH secretion be reassessed in all patients after withdrawal of the growth-promoting therapy and to identify ones with persistent severe GHD [1]. Taking into account the lower cut-off values for the GH peak in stimulation tests confirming GHD in retesting than that established for children, as well as the observed normalization of previously decreased GH secretion in the majority of patients with isolated, idiopathic GHD, diagnosed in childhood, only a small proportion of them should continue rhGH therapy during adult life. A very important problem is determining the moment at which this retesting should be performed.

In the Consensus of the GH Research Society (GHRS) from 2000 [3], it was recommended that GH stimulation tests be repeated after the attainment of FH and that severe GHD be diagnosed in the case of a GH peak below 3.0 µg/L in an insulin tolerance test (ITT). This threshold seems too low for patients in the transition period. In 2005, Clayton et al. [4] proposed increasing the cut-off value for patients in late puberty up to 5.0 µg/L. Two years later, in the Consensus of the GHRS in cooperation with Endocrine Societies from Europe, US, Japan and Australia [5], a cut-off value for the GH peak in stimulation tests diagnostic for GHD during the transition period was established at the level of 6.0 µg/L. According to recent American guidelines [6], patients with a diagnosis of isolated, idiopathic GHD who have an insulin-like growth factor-1 (IGF-1) SDS above 0.0 after rhGH therapy withdrawal and the wash-out period do not require retesting to exclude persistent GHD.

In 2016, Grimberg et al. [1] strongly recommended that rhGH treatment at pediatric doses should not be continued after the attainment of near-FH, defined as a height velocity (HV) of below 2–2.5 cm/year in boys with a bone age (BA) of 16–17 years and in girls with a BA of 14–15 years. All the patients who fulfilled these criteria should be reassessed for GH secretion, as rhGH therapy should be resumed only in the case of confirmed persistent severe GHD. More recent studies by Penta et al. [7] and Cavarzere et al. [8] have shown that normalization of GH secretion in children with isolated, idiopathic GHD may occur in mid-puberty, before the attainment of near-FH. Moreover, already in 2002, Loche et al. [9] reported the possibility of an even earlier normalization of GH secretion in children previously diagnosed with GHD.

According to the considerations of Bright et al. [10], the rate of false-positive results in GH stimulation tests, resulting in false-positive diagnoses of GHD, may be higher (even much higher) than the rate of true positives. The authors have stressed the need to improve the diagnostic scheme due to the “low probability of GHD in a child with short stature and positive growth hormone stimulation tests”. The same problem with respect to children with idiopathic, isolated GHD has been discussed in a recent paper by Allen [11]. Such a situation is suggested to be an important cause of the overdiagnosis of GHD, leading to the overtreatment of short children with rhGH. Moreover, the phenomenon of “transient” GHD may also be considered a falsely positive diagnosis of GHD rather than evidence of the normalization of previously decreased GH secretion. However, there are no diagnostic tools to clearly evaluate in which patients this could occur.

Insulin-like growth factor-1 is the main peripheral mediator of GH action. In the classification of pediatric endocrine diagnoses, provided by Wit et al. in 2007 [12], GHD was described as a synonym for secondary IGF-1 deficiency. Nevertheless, in the same year, GHRS published an updated Consensus concerning GHD in adults, in which the authors clearly stated that, in adults, a normal IGF-1 concentration did not exclude GHD [5]. The same statement was confirmed by other authors [13,14]. However, according to more recent guidelines, the diagnosis of GHD seems unlikely in patients with an IGF-1 concentration over the median value in the population (i.e., IGF-1 SDS over 0.0), and patients with a diagnosis of isolated, idiopathic GHD in childhood and an IGF-1 SDS above 0.0 after rhGH therapy withdrawal and the wash-out period do not require retesting to exclude persistent GHD [6,15].

In 2020, Wit et al. [16] proposed the use of serum IGF-1 concentration as part of the screening in children with growth failure, with different cut-offs, established with respect to the clinical likelihood of GHD. The authors decided to introduce IGF-1 SDS for the BA and the pubertal stage, not only for the chronological age (CA) of children.

In Poland, during last 20 years, different criteria of rhGH therapy withdrawal have been applied to children diagnosed with GHD. Initially, rhGH therapy was discontinued when the child attained a height corresponding to the value of 10th centile for the age of 18 years (170 cm for boys and 158 cm for girls), according to Polish reference charts [17], or a BA equal or above 16 years in boys and 14 years in girls, with the possibility of the earlier cessation of ineffective treatment if the patient’s HV was below 3 cm/year. Next, when the availability of rhGH improved and the knowledge of its metabolic effects increased, the period of treatment was extended up to the attainment of a BA above 18 years for boys and 16 years for girls, independently of the patient’s height and current HV; the recommendation for its earlier termination due to an HV below 3 cm/year still remained, but only in patients with a BA below 16 years for boys and 14 years for girls. The latter criteria were modified according to the guidelines of Grimberg et al. [1] in mid-2022, i.e., after closing the database for the present study. However, in practice, some patients who attained a satisfactory height and/or who experienced a significant slowdown in their HV decided to stop treatment before fulfilling the withdrawal criteria. This was especially true for boys with a BA over 16 years and for girls with BA over 14 years. Unfortunately, some of them did so without consulting a doctor, thus becoming lost to follow-up. All the patients remaining under observation were asked to be subjected for retesting after rhGH therapy discontinuation and the wash-out period. This re-evaluation of GH secretory status included at least one GH stimulation test in each patient and measurement of the IGF-1 concentration. Thus, we have been able to collect retrospective data from a cohort of patients diagnosed with GHD in childhood who had completed growth-promoting therapy and who had performed retesting.

The aim of the present study was to assess the incidence of persistent and transient GHD and the effectiveness of rhGH therapy in the patients diagnosed in childhood with isolated, idiopathic GHD, with respect to the moment of rhGH therapy withdrawal (before or after the attainment of near-FH), according to different diagnostic criteria of GHD for childhood and for the transition period. Taking into account the greater number of boys than girls qualified for rhGH therapy, as well as the gender-related criteria of rhGH therapy withdrawal, the patients were also divided by sex during the analysis.

## 2. Results

### 2.1. Assessment of the Incidence of Transient GHD with Respect to Pediatric Criteria from 2000 (GH Peak below 10 µg/L, IGF-1 Deficiency Not Required) [3] in Relation to the Moment of rhGH Therapy Withdrawal and Patients’ Sex

Detailed characteristics, including pre-treatment data, effectiveness of rhGH therapy and hormonal findings upon retesting, for the whole studied group and for the subgroups who completed treatment either before (Group PRE) or after (Group POST) the attainment of near-FH is presented in Table 1. At the onset of rhGH therapy, there was no significant difference between Groups PRE and POST in their height SDS at therapy onset (hSDS-0) and in the results of hormonal tests (GH peak and IGF-1 SDS). There were also no differences in both the GH peak and IGF-1 SDS at retesting between these groups.

The increase in height SDS at retesting (hSDS-R) with respect to hSDS-0 (Δ hSDS) was significant (*p* < 0.001) in both groups. During retesting, the GH peak and IGF-1 SDS were significantly (*p* < 0.001) higher than before treatment in each group; moreover, the median GH peak and median IGF-1 SDS were normal in both groups (see Figure 1).

Despite a significantly (*p* < 0.001) shorter duration of rhGH therapy in Group PRE than in Group POST, both the hSDS-R and Δ hSDS were significantly (*p* < 0.001) higher in Group PRE than in Group POST. Nevertheless, the hSDS-R was very close to the target height (TH) SDS in both groups, with no significant difference between PRE and POST (for details, see Table 1).

Normalization of GH secretion at retesting with respect to the criteria for children (i.e., GH peak over 10.0 µg/L) was observed in 168 out of 260 patients (64.6%); 58 patients had GHD only with respect to the pediatric criteria (i.e., GH peak between 6.0 and 10.0 µg/L), while only 34 patients (13.1%) had persistent GHD according to the criteria for transition period (i.e., GH peak below 6.0 µg/L), including only 10 patients (3.8%) fulfilling the current criteria for persistent severe GHD (i.e., GH peak below 3.0 µg/L). There was no significant difference in the frequency of normal GH peaks at retesting according to the pediatric criteria between Groups PRE and POST. This observation indicates that normalization of GH secretion took place before the attainment of near-FH. Detailed data with respect to the different cut-offs for the GH peak at retesting are provided in Table 2.

Despite this, the frequency of transient GHD with respect to the pediatric criteria was significantly (*p* = 0.008) higher in boys than in girls. Detailed data with respect to different cut-offs for the GH peak at retesting and patients’ sex are provided in Table 3.

The only differences between boys and girls, both before treatment and at retesting, were related to the patients’ age (significantly younger in girls; *p* < 0.001 at both timepoints), the GH peak at retesting and the Δ GH peak (significantly higher in boys; *p* < 0.001) and the Δ hSDS (significantly better in boys), while there was no significant difference in other analyzed auxological indices (for details, see Table 4). The increases in the GH peak and the IGF-1 SDS at retesting with respect to pre-treatment values, as well as in hSDS-R with respect to hSDS-0, were significant (*p* < 0.001), both in boys and in girls.

The presented analyses have shown the predominance of boys in the group of treated children and with a high incidence of transient GHD, even with respect to the criteria for childhood and even if retesting was performed before the attainment of near-FH, especially in boys. Effectiveness of the rhGH therapy was greater in patients who stopped treatment before the attainment of near-FH (which may be a kind of bias as some patients who did not respond well could be treated longer than good responders who gave up treatment after the attainment of a satisfactory height). Thus, the first important conclusion from our study could be the statement that normalization of GH secretion in stimulation tests occurred in the majority of patients with childhood-onset isolated, idiopathic GHD and took place before the attainment of near-FH. Moreover, cessation of rhGH therapy before the attainment of near-FH appeared to have no adverse effect on its effectiveness.

The overall high frequency of transient GHD, which was even higher in boys than in girls, has suggested the possibility of the overdiagnosis of GHD (and overtreatment with rhGH) in a significant proportion of the patients, especially in boys.

Thus, we decided to check if applying more restricted diagnostic criteria of GHD, as proposed in recent years [15], could be useful in overcoming these problems.

### 2.2. Verification of GHD Diagnosis in Childhood and the Incidence of Transient GHD with Respect to Currently Proposed Pediatric Criteria [15]

For the next part of the analysis, the following three subgroups of patients were extracted:✓“Unlikely” GHD—patients with an IGF-1 SDS above 0.0, who could have GHD excluded without stimulation tests;✓“Doubtful” GHD—patients with an IGF-1 SDS below 0.0 and a GH peak in stimulation tests of between 7.0 µg/L and 10.0 µg/L, who do not fulfill the criteria for GHD after applying a lower cut-off for the GH peak in stimulation tests;✓“True” GHD—patients with an IGF-1 SDS below 0.0 and a GH peak of less than 7.0 µg/L; the only subgroup that should be treated with rhGH, according to current guidelines [15].

Consequently, at retesting, we decided to qualify as non-GH-deficient all the patients with an IGF-1 SDS above 0.0 (regardless of the results of the GH stimulation tests) and to use cut-off values of 7.0 µg/L and 5.0 µg/L for the GH peak in patients with an IGF-1 SDS below 0.0. Thus, we divided the patients with respect to the criteria for the transition period into the following groups:✓transient GHD (Group transGHD)—patients with an IGF-1 SDS at retesting above 0.0 or a GH peak at retesting above 7.0 µg/L;✓“childhood-only” GHD (Group childGHD)—patients with an IGF-1 SDS below 0.0 and a GH peak of 5.0–7.0 µg/L;✓persistent GHD (Group persistGHD)—patients with an IGF-1 SDS below 0.0 and a GH peak at retesting below 5.0 µg/L.

Transient GHD with respect to pre-treatment criteria for children was observed in 89.1% of children with “Unlikely” GHD before treatment, 89.6% of those with “Doubtful” GHD and as much as 86.2% of those diagnosed with “True” GHD. Thus, changes to the diagnostic criteria for GHD in childhood could decrease the number of patients qualifying for rhGH therapy but not the incidence of transient GHD at retesting with respect to pre-treatment criteria. The number of patients in the individual groups and subgroups are presented in Table 5.

Significant differences in the effectiveness of rhGH therapy between the groups with “Unlikely”, “Doubtful” and “True” GHD were observed. Post hoc analysis showed that the hSDS-R was significantly (*p* < 0.001) lower in the patients with “Unlikely” GHD than in the remaining groups (the same was seen for the corr hSDS-R), while Δ hSDS was significantly (*p* = 0.010) higher in “Doubtful” GHD than in other groups. Nevertheless, hSDS-R values exceeded those of hSDS-0 significantly (*p* < 0.001) in all groups. In the group with “Unlikely” GHD, the lower quartile of hSDS-R had values below −2.0, which means that over 25% of patients from this group had short stature at retesting.

Despite pre-treatment differences in the IGF-1 SDS, defining the assignment of patients to the “Unlikely” GHD group and to the remaining groups (“Doubtful” and “True” GHD), the differences in the IGF-1 SDS at retesting between these groups were insignificant. However, the IGF-1 SDS increased in the “True” GHD and “Doubtful GHD groups, while it decreased in the “Unlikely” GHD group. The differences in Δ IGF-1 SDS between the group with “Unlikely” GHD and the remaining groups (“Doubtful” and “True” GHD) were significant (*p* < 0.001), but there was no significant difference between the “True” GHD and “Doubtful” GHD groups. For details, see Table 6 and Figure 2.

### 2.3. Characteristics of Patients with Persistent Severe GHD According to Current Criteria (GH Peak at Retesting below 3.0 µg/L) [18]

Only 10 out of 260 patients (3.85%), including 7 out of 173 boys (4.05%) and 3 out of 87 girls (3.45%), had a GH peak below 3.0 µg/L at retesting, including 7 patients (5 boys and 2 girls) with an IGF-1 SDS below −2.0 that additionally confirmed the diagnosis of severe GHD. Among these seven patients, five were diagnosed in childhood with “True” GHD, while one boy was diagnosed with “Unlikely” GHD and one girl with “Doubtful” GHD. This boy was the youngest patient in the whole group (3.5 years old at diagnosis, 3.7 years old at rhGH therapy onset), with a very severe height deficit (hSDS −5.98) and a very low GH peak (1.2 µg/L) but “normal” IGF-1 SDS. The patient was diagnosed in 1997 and had his IGF-1 measured with the RIA method, which was used for a relatively small number of patients and has not been used in our center for 20 years, so it is difficult to explain the situation. The girl was aged 11.7 years at diagnosis; she had a severe height deficit (hSDS −3.13), a GH peak of 7.43 µg/L and a severe IGF-1 deficiency (IGF-1 SDS −2.11) that persisted at retesting (IGF-1 SDS −2.57). She attained an FH SDS of −1.47, thus improving her height by 1.66 SD with respect to her hSDS at diagnosis. Among the remaining three patients with a GH peak at retesting of below 3.0 µg/L, two boys had relatively high IGF-1 SDSs (0.83 and 1.03) and one girl had a “borderline” IGF-1 SDS (0.09), which could suggest falsely positive results from the GH stimulation tests during their retesting; among them, the girl and one boy had a pre-treatment IGF-1 SDS above 0.0, while the second boy had a GH peak of 2.72 ng/mL and an IGF-1 SDS below −2.0 at diagnosis.

### 2.4. Effect of the Application of the Diagnostic Strategy Proposed by Wit et al. [16] for the Diagnosis of GHD in Childhood and during Retesting

According to the proposal of Wit et al. [16], IGF-1 assessment should be a part of laboratory screening for GHD to assess the pre-test likelihood of the disease. This approach assumes subjecting to GH stimulation tests only the patients with a real likelihood of GHD, based on the clinical assessment and IGF-1 concentration, interpreted with respect to BA or pubertal stage. According to the proposed criteria and taking into account the inclusion criteria for the current study, our patients had a low, very low or rather low pre-test likelihood of GHD, so the IGF-1 SDS cut-off for them should be −2.0 or −1.0. In the retrospective analysis, it was difficult to clearly distinguish patients between these three subgroups; nevertheless, none of our patients had a moderate or high pre-test likelihood of GHD. Unfortunately, for some patients, exact data concerning the pubertal stage was missing or reported as intermediate between subsequent Tanner stages. So, we decided to calculate the IGF-1 SDS with respect to BA and to apply the cut-off for the IGF-1 SDS at the level of −1.0. Thus, all patients with an IGF-1 SDS for BA of over −1.0 were assigned, independently from the results of GH stimulation tests, to the “Pre-test Unlikely” GHD group (which is not the same as the “Unlikely” group in the previous analysis), while the remaining ones were qualified for the groups with “Excluded” or “Confirmed” GHD, depending on their GH peak in stimulation tests (below or over 7.0 µg/L). At retesting, “Transient” GHD (T-GHD) was defined as having an IGF-1 SDS of over −1.0 or a GH peak above 7.0 µg/L and “Non-transient” GHD (NT-GHD) as having an IGF-I SDS below −1.0 and a GH peak between 5.0 and 7.0 µg/L, while “Persistent” GHD (P-GHD) was defined as having an IGF-I SDS below −1.0 and a GH peak below 5.0 µg/L (please note that this is not the same group as persistGHD in the previous analysis).

As a result, 168 patients (64.6%) were qualified to the “Pre-test Unlikely” group, 45 (13.5%) were recognized as those with “Excluded” GHD, while only 47 patients (18.1%) were diagnosed with “Confirmed GHD”. At retesting, only 50 patients had an IGF-1 SDS below −1.0 and only 13 of them (9 boys, 4 girls) had a GH peak at retesting of below 7.0 µg/L; among them, only 3 patients had a GH peak between 5.0 and 7.0 µg/L. The incidence of “Transient” GHD with respect to pre-treatment criteria was still very high: 87.2% in the “Confirmed” GHD group, 93.3% in the “Excluded” GHD group and 97.6% in the “Pre-test Unlikely” GHD group. Unfortunately, according to the proposed criteria, only 6 out of 13 patients fulfilling the pre-treatment criteria for GHD at retesting would have qualified as ones with “Confirmed” GHD and be subjected to rhGH therapy, while the remaining 7 would have remained untreated (for details, see Table 7). So, applying more restrictive criteria for diagnosing GHD might significantly decrease the number of treated patients; however, after applying the same criteria at retesting, the incidence of transient GHD remained very high.

The effectiveness of rhGH therapy was lower in the patients with “Pre-test Unlikely” GHD than in the remaining groups (“Excluded” and “Confirmed” GHD), as post hoc analysis showed that the hSDS-R was significantly lower in “Pre-test Unlikely” GHD than in both “Excluded” GHD (*p* = 0.006) and “Confirmed” GHD (*p* = 0.03); the same was true for the corresponding differences in Δ hSDS (*p* = 0.018 and *p* = 0.013, respectively). There was no difference in any of the auxological indices of rhGH therapy effectiveness (hSDS-R, corr hSDS-R, Δ hSDS) between the patients with “Excluded” and “Confirmed” GHD, and the only differences between these two groups concerned the results of the GH stimulation test. Despite the fact that the GH peak at retesting was significantly lower in the group with “Confirmed” GHD than in both the “Excluded” GHD (*p* < 0.001) and “Pre-test Unlikely” GHD (*p* = 0.16) groups, the median value of the GH peak at retesting was over the cut-off for diagnosis in all groups. Moreover, the value of the lower quartile of the GH peak at retesting did not exceed this cut-off only in “Confirmed” GHD. Nevertheless, as both groups were treated with rhGH, it is very difficult to speculate if one of these groups should be treated, while the second could really remain untreated. Detailed data are presented in Table 8.

### 2.5. Characteristics of Poor Responders to rhGH Therapy

At retesting, 44 patients with a Δ hSDS below 1.0 fulfilled the criteria for a poor response to rhGH therapy. This was not consistent with the number of patients who did not reach “normal” height as, in some patients, their hSDS at retesting remained below −2.0 despite its significant improvement with respect to the pre-treatment value. Thus, non-responders constituted 16.92% of the entire group. It seems important that over one third (33.36%) of patients with “Unlikely” GHD, while only every tenth with “Doubtful” GHD (10.42%) and every eighth with “True” GHD (12.84%), classified according to the criteria of GHRS from 2019 [15], turned out to be non-responders.

Especially important seems to be the distribution of poor responders among the groups, extracted according to the criteria proposed by Wit et al. [16], as only 5 of them were in the “Confirmed” GHD group (10.64% of the group), while 3 were in the “Excluded” GHD group (6.67% of the group) and the remaining 36 were in the “Pre-test Unlikely” GHD group (28.13% of the group). Moreover, at retesting, all poor responders had transient GHD with respect to the pre-treatment criteria of Wit et al. [16].

## 3. Discussion

Previous studies conducted in the last decade of the 20th century showed that a significant proportion of patients with GHD diagnosed in childhood had sufficient GH secretion during retesting after completion of linear growth, especially if the GH deficit was idiopathic and not associated with other disorders of the pituitary function [19,20,21]. The results of other studies pointed to the possibility of an earlier normalization of GH secretion, raising questions concerning the optimal duration of rhGH therapy. In 1992, Cacciari et al. [22] observed the normalization of previously decreased GH secretion 1 month after the withdrawal of rhGH therapy in the majority of pubertal children but in none of those who were still prepubertal at retesting, which indicated that some children diagnosed as GH-deficient did not have a true GHD. Ten years later, Loche et al. [9] reported normal GH responses to pharmacological stimulation in 28 out of 33 prepubertal children with normal magnetic resonance imaging (MRI) of the pituitary region, who had been diagnosed with GHD on the grounds of the same stimulation tests just 1–6 months earlier. The authors pointed out the need to follow-up and reassess such patients before a diagnosis of GHD was definitely established. In 2006, Zucchini et al. [23] reported normal GH secretion at puberty in one third of children diagnosed with isolated GHD before puberty. Moreover, despite the withdrawal of rhGH therapy in this group of children, the attained adult height was similar among them and in the remaining patients, who completed treatment when their HV slowed down to below 1 cm/year, and the BA was at least 17 years in boys and 15 years in girls. In their paper published in 2015, Bizzarri et al. [24] demonstrated the normalization of GH secretion at retesting performed during puberty or even before its onset in 36 out of 38 children with a previous diagnosis of idiopathic GHD; in the remaining 2 subjects, the GH peak was decreased but IGF-1 levels were normal. The authors pointed out the high rate of falsely positive results from GH stimulation tests and the need for early retesting in the case of an unsatisfactory response to GH therapy. Five years ago, Penta et al. [7] reported persistent GHD (but with normal IGF-1 levels) in the transition period in only 5 out of 31 patients diagnosed with idiopathic GHD in childhood. In conclusion, they suggested that the timing of retesting should be anticipated in order to avoid overtreatment. In 2020, Cavarzere et al. [8] documented normal GH secretion at retesting after the wash-out period in 55% of children in the intermediate stage of puberty who had been diagnosed previously with GHD and treated with GH for at least 2 years. Similarly to previous observations of Zucchini et al. [23], definitive discontinuation of treatment in this group of patients did not impair their growth potential (their adult height was similar as in the remaining 45% of children with confirmed GHD, treated up to the attainment of near-FH).

On the other hand, Lanzetta et al. [25] have reported similar effectiveness of rhGH therapy in the patients with definite GHD (confirmed genetic defect, combined pituitary hormone deficiency, anatomical hypothalamic–pituitary abnormality) and with “short stature unresponsive to stimulation” (decreased GH peak in stimulation tests but no identified cause of impaired GH secretion).

In our previous study [26], the incidence of persistent GHD (with the cut-off limit for GH peak in two stimulation tests appropriate for the transition period, i.e., 6 µg/L) was only 12% in patients diagnosed with isolated, non-acquired GHD in childhood who had completed growth-promoting treatment and attained their FH.

In the already quoted paper, Cacciari et al. [22] stated that the observed discrepancies between the results of repeated GH stimulation tests were caused by errors in the diagnosis of GHD, including variability in the GH response to stimulation (pharmacological or physiological), transient GHD and the difficulties in distinguishing true GHD from delayed puberty.

It seems very important to recognize if we are dealing with a real (true) normalization of GH secretion at puberty in children who were actually GH-deficient before puberty or with poor reproducibility of the results of GH stimulation tests, manifesting as apparent (false) GHD that disappears at the time of retesting.

The incidence of transient GHD with respect to pre-treatment criteria among our patients was high and independent from the moment of retesting (before or after the attainment of near-FH), which confirms that GH secretion appears normal earlier than when the patients fulfill the current exclusion criteria for rhGH treatment. Unfortunately, applying more restrictive diagnostic criteria in childhood may decrease the number of patients who qualify for rhGH therapy but not the incidence of transient GHD if the same criteria are used for retesting, which indicates that the problem of overdiagnosing childhood-onset idiopathic, isolated GHD may be only partially resolved by such modifications.

In our study, similarly as in the majority of papers concerning rhGH therapy in children with GHD, the number of boys exceeded the number of girls, suggesting the higher incidence of GHD among boys than among girls. However, according to the report by Tanaka et al. [27], this difference seems to reflect a selection bias related to a higher proportion of boys among children seeking treatment due to their short stature. Interestingly, the incidence of transient GHD was higher among boys than among girls, with a similar number of male and female patients fulfilling the criteria for persistent GHD. Our study was limited to the patients with isolated GHD; nevertheless, the obtained results seem to be in line with the very recent report by Henry et al. [28], who observed male predominance in referrals to clinic for short stature, but not among patients with an abnormal pituitary MRI and/or multiple pituitary hormone deficiency. It seems worth considering that the real incidence of GHD may be definitely lower than the frequency of diagnosing this disease in children and independent from the patients’ sex.

We observed a wide range of growth responses to rhGH therapy, with the highest rate of poor responders among the patients with “Pre-test Unlikely” GHD, according to the criteria proposed by Wit et al. [16]. Moreover, with respect to the same criteria, none of the poor responders had persistent GHD. These findings suggest the possibility of other causes of short stature in a significant proportion of poor responders, who were all ineffectively treated as GH-deficient. It seems impossible to forecast what would happen if the modified diagnostic criteria were applied to our patients. Nevertheless, our results confirm the need for further research on the diagnostic criteria for idiopathic, isolated GHD in children and on the optimal rhGH therapy duration.

One direction of research should be the introduction of prediction models of rhGH therapy effectiveness into clinical practice in order to have reasonable expectations about the effects of treatment and to identify poor responders as early as possible. Prediction models have been developed for many years with the use of different methods, like linear and non-linear regression, neural networks or other advanced computational techniques; however, only few of them allow the therapy outcome to be predicted [29,30,31,32,33,34]. It seems that the availability of such models in the form of easy-to-use calculators would facilitate the decision to discontinue GH therapy and to perform retesting earlier.

The second direction is the better identification of the moment of normalization of GH secretion in order to discontinue treatment in patients in whom GHD is transient. In poor responders, extended diagnostics should be considered, especially in the case of transient GHD, as the real cause of short stature in them might remain unrecognized. In this aspect, the recent project of Brettell et al. [35], who decided to assess the effect on FH of the discontinuation versus continuation of rhGH therapy in pubertal children with isolated GHD, seems to be very important.

Last, but not least, it seems that the high incidence of falsely positive results of GH stimulation tests in children with short stature should be taken into account. According to previous studies, this problem may concern a relatively high proportion of patients [10,11,16,36].

The use of sex-steroid priming in selected groups of children has been proposed, and this was performed in Poland many years ago. The patients subjected to the present study did not have such priming due to the lack of clear national recommendations at the moment of their diagnosis. Priming has been recommended in the guidelines of Grimberg et al. [1] and in more recent papers [15,37,38,39]; thus, it is a potential solution to decrease the number of the patients overdiagnosed with GHD. It seems possible that the problem of falsely positive results of GH stimulation tests may—at least to some extent—concern retesting; however, there is no sufficient data on this issue. This could explain the diagnosis of persistent GHD at retesting in some patients who did not fulfill the current diagnostic criteria of this disease in childhood; however, this is more of an assumption than a conclusion.

Another topic for further research should be the revision of the criteria for rhGH therapy effectiveness in children diagnosed with isolated, idiopathic GHD in the context of the exclusion of a previous diagnosis in the majority of treated patients. If some of them were not GH-deficient during the entire therapy duration or were even falsely diagnosed as GH-deficient, it is possible that some of the apparently good responders could grow well without treatment. Such an eventuality has been suggested in a recent paper by Bright et al. [10]. On the other hand, in our study, the highest proportion of poor responders was among the patients who should have been classified into the “Excluded” or “Unlikely” GHD groups according to the current criteria. Labelling such patients as GH-deficient might lead to the failure to complete diagnostic tests for other causes of short stature. Indeed, in recent years, the possibility of revealing genetic defects in children previously treated as having idiopathic short stature, has significantly expanded [40]. Until the optimal criteria for the diagnosis of childhood isolated, idiopathic GHD are established, earlier retesting of poor responders to rhGH therapy, followed by extended genetic diagnostic tests in the case of unconfirmed GHD, seems worth considering as an alternative to long-term ineffective treatment.

## 4. Materials and Methods

The retrospective analysis included 260 patients (173 boys, 87 girls) with short stature who fulfilled the diagnostic criteria for isolated, idiopathic GHD and had been treated with rhGH in childhood, completed growth-promoting therapy and had had their GH secretory status reassessed after the wash-out period. Patients with other hormonal deficits (except for well-controlled primary hypothyroidism), malnutrition, chronic diseases, neoplastic processes, diagnosed or suspected skeletal dysplasia, long-term or recurrent glucocorticoid administration and other therapies that might disturb growth or the function of somatotropic axis were excluded from the study. Pubertal development was spontaneous and completed at therapy withdrawal in each case; however, it could have been delayed. All the girls included had a normal female karyotype (46, XX). Magnetic resonance imaging of the hypothalamic–pituitary region was performed in each case before treatment, and all the patients with pituitary–hypothalamic structural defects (except for isolated decreased size of the anterior pituitary) and/or with a history of brain injuries, neurosurgery or tumors were excluded from the study. The diagnosis of GHD in childhood was based on auxological and hormonal criteria.

### 4.1. Auxological Indices

Auxological criteria included the following: short stature below the 3rd centile according to national growth charts [17]; slow HV, expressed as a decrease in the patient’s hSDS during at least 6 months of observation; and a delayed BA.

For 224 patients, data concerning the heights of their parents were available, and the TH SDSs were calculated. Patients’ corrected height SDS (corr hSDS) at diagnosis were calculated as the difference between hSDS-0 and the TH SDS.

During rhGH treatment, the patients’ BA was routinely assessed at least once a year, including the last X-ray at therapy withdrawal. For each patient, near-FH was defined in concordance with the recommendations of Grimberg et al. [1] as achieving a BA of over 16 years in boys and over 14 years in girls, together with a deceleration in HV of below 2.5 cm/year (based on measurements at least 6 months apart). That allowed the classification of patients with respect to the moment of rhGH therapy withdrawal into Group PRE—patients who had completed treatment before the attainment of near-FH—or Group POST—patients who were treated up to the attainment of near-FH or for longer. The patients in whom the advancement of BA had not fully corresponded to the growth rate were not included in the study due to the difficulties in clearly determining the moment of the attainment of near-FH.

At retesting, patients’ heights were measured, and the value of hSDS-R for age and sex were calculated and compared with the hSDS-0 (Δ hSDS was calculated as the difference between hSDS-R and hSDS-0). The patients with a Δ hSDS below 1.0 were classified as poor responders to rhGH therapy. The difference between hSDS-R and TH SDS (corr hSDS-R) was calculated as an additional index of rhGH therapy effectiveness.

All values of hSDS-0, hSDS-R and TH SDS were calculated according to the Polish reference charts of Palczewska and Niedźwiecka [17]. Bone age was assessed according to Greulich–Pyle standards [41].

### 4.2. Hormonal Tests

Hormonal confirmation of GHD in childhood was a GH peak below 10.0 µg/L in two stimulation tests: with clonidine 0.15 mg/m^2^ orally (GH concentrations measured every 30 min from 0 to 120 min in the test) and with glucagon 0.03 mg/kg (not exceeding 1.0 mg) intramuscularly (GH concentrations measured at 0, 90, 120, 150 and 180 min in the test). However, taking into account the suggestions from more recent studies [15,42,43] that this threshold might be too high, for several analyses, a subgroup of children with a GH peak in both tests of below 7.0 µg/L was extracted.

Reassessment of GH secretion (retesting) was performed after the completion of rhGH therapy and a wash-out period (between 1 and 6 months). All the patients had performed an ITT with insulin at a dose 0.1 IU/kg, given intravenously (GH and glucose concentrations measured every 30 min from 0 to 120 min of test; the test had to be repeated if there was no glucose decrease below 40.0 mg/dL or by at least 50% of initial value), and the second test with clonidine, with the drug dose and timepoints of GH sampling as in childhood (this test could be withdrawn in patients with a GH peak in the ITT of over 10.0 µg/L).

The three cut-off values were applied in the first statistical analysis: 3.0 µg/L (cut-off value for diagnosing persistent severe GHD in adults) 6.0 µg/L (cut-off value suggested for patients in the transition period) and 10.0 µg/L as a standard threshold for diagnosing GHD in children. Next, we chose lower thresholds for the GH peak in retesting: 5.0 µg/L (according to the most recent guidelines) [6] and 7.0 µg/L (the same value as recently proposed for children). Using the same cut-offs for the GH peak at retesting as in childhood seemed justified as the analysis included the patients who stopped treatment before the attainment of near-FH.

The concentrations of GH were measured by the two-site chemiluminescent enzyme immunometric assay (hGH IMMULITE, DPC) for the quantitative measurement of human GH, calibrated to the WHO IRP 80/505 standard [44] or to the 98/574 standard [45].

In all patients, IGF-1 concentrations were assessed both before treatment and during retesting. In the years 2003–2016, IGF-1 concentrations were measured using solid-phase, enzyme-labelled chemiluminescent immunometric assays (IMMULITE, DPC), calibrated to the WHO NIBSC 1st IRR 87/518. Since 2017, new IGF-1 assays, standardized to the WHO 1st International Standard 02/254, have become available in Poland, which resulted in the need to convert the obtained results according to the equation provided in Siemens Healthcare Diagnostics Customer Bulletin for IMMULITE^®^ 2000 Immunoassay System, from May 2016 [46]. The agreement in IGF-1 measurements by different assays, especially for lower limits of reference values and with differences in raw values but not in SDS values, has been confirmed in dedicated studies [47,48]. All IGF-1 values were expressed as IGF-1 SDS for age and sex, according to the formula proposed by Blum and Schweitzer [49] and reference data for the laboratory method used [50]. Up to 2003, IGF-1 concentrations were measured with IGF-I-D-RIA-CT (Biosource Europe, Belgium); this method was used only for 74 patients in the pre-treatment period, for these measurements, IGF-1 SDS values were calculated according to the reference data provided with the kits.

For the last analysis, the IGF-1 SDS for BA was calculated according to the same rules as for CA (which was clearly indicated); everywhere else, IGF-1 SDS values refer to CA.

It should be noted that IGF-1 deficiency was not listed among the criteria to qualify patients for rhGH therapy in Poland; thus, some children diagnosed with GHD could have normal IGF-1 levels before treatment, including ones with an IGF-1 SDS above 0.0.

Taking into account the rule that isolated, idiopathic GHD is unlikely in patients with an IGF-1 SDS above 0.0 [6,15], this cut-off value was applied in some analyses. Finally, the criteria of pre-test likelihood of GHD proposed by Wit et al. [16], based on clinical positive and negative clues, and IGF-1 concentration, were retrospectively applied to our cohort of patients.

### 4.3. Statistical Analysis

Statistical analysis was performed for the subgroups of patients, divided according to the moment of rhGH therapy withdrawal and sex, as well as with respect to different cut-offs for the GH peak in stimulation tests and the IGF-1 SDS at diagnosis and during retesting.

First, the distributions of the analyzed variables were assessed by the Shapiro–Wilk test. As they usually did not follow a normal distribution in at least one group, descriptive statistics reported the median and interquartile ranges of the variables. The comparisons between the groups of patients were performed with nonparametric tests: the Mann–Whitney U test was selected for comparisons of continuous variables between two groups, while the Kruskal–Wallis ANOVA with appropriate post hoc tests was used for more than two groups. Two measurements of the same variable in the same group at different time points were compared by the Wilcoxon test; the t test for repeated measurements was not chosen since the normality assumption was often violated. Fisher’s exact test was used for the assessment of differences in the frequency of persistent and transient GHD, as well as for the proportions of males and females in particular groups. Analysis was performed with Statistica 13.3.

## 5. Conclusions

Normalization of GH secretion in patients diagnosed at childhood with idiopathic, isolated GHD is observed in most patients and occurs before the attainment of near-final height.

Applying more restrictive diagnostic criteria for GHD in childhood (the necessity of confirming an IGF-1 deficiency, lower cut-offs for the GH peak in stimulation tests) may decrease the number of patients who qualify for rhGH therapy, but not the incidence of transient GHD among treated patients, if the same pre-treatment criteria are applied during retesting.

Further efforts for optimizing the diagnosis of isolated, idiopathic GHD and the rhGH therapy duration should be made in order to avoid overdiagnosis and overtreatment of patients with only apparent disease and to rationalize treatment costs.

Special attention should be paid to poor responders to rhGH therapy for whom the diagnosis of GHD may be a false positive and for whom the real cause of short stature remains undiagnosed.

## Figures and Tables

**Figure 1 ijms-25-05739-f001:**
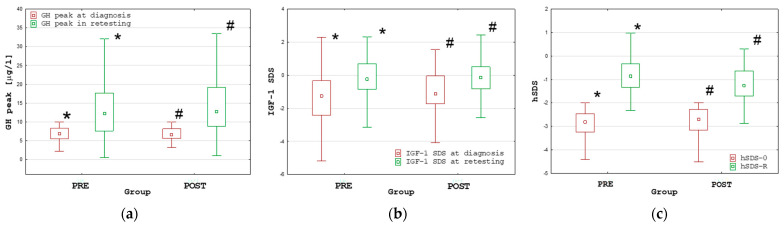
Comparisons of the GH peak (**a**), IGF-1 SDS (**b**) and height SDS (**c**) at diagnosis and during retesting in the PRE and POST groups; all values are presented as the median (point), interquartile range (box) and non-outlier range (whiskers); significant differences (*p* < 0.001), according to the Wilcoxon test, are marked with * and #.

**Figure 2 ijms-25-05739-f002:**
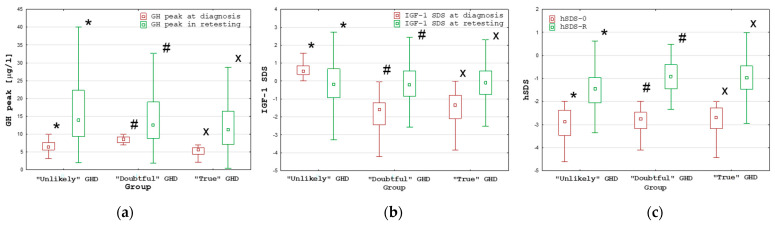
Comparisons of the GH peak (**a**), IGF-1 SDS (**b**) and hSDS (**c**) at diagnosis and during retesting between the groups with “Unlikely, “Doubtful” and “True” GHD; all values are presented as the median (point), interquartile range (box) and non-outlier range (whiskers); significant differences (*p* < 0.001) as measured by the Wilcoxon test are marked with *, # and x.

**Table 1 ijms-25-05739-t001:** Pre-treatment characteristics, indices of rhGH therapy effectiveness and hormonal status at retesting of patients who completed rhGH therapy before (Group PRE) or after (Group POST) the attainment of near-FH.

	All	Group PRE	Group POST	*p*
N (boys/girls) *	260 (173/87)	121 (92/29)	139 (81/58)	
Age at therapy onset [years]	13.2 (11.9; 14.4)	13.7 (12.4; 14.7)	12.7 (11.5; 14.0)	**0.001**
Age at retesting [years]	17.6 (16.5; 18.3)	17.6 (16.8; 18.3)	17.6 (16.4; 18.3)	0.985
Therapy duration [years]	4.2 (3.4; 5.2)	3.9 (2.9; 4.7)	4.5 (3.8; 5.6)	**<0.001**
GH peak at diagnosis [µg/L]	6.80 (5.58; 8.23)	6.86 (5.51; 8.30)	7.22 (0.90; 10.70)	0.960
GH peak at retesting [µg/L]	12.40 (7.98; 18.45)	12.20 (7.57; 17.70)	12.80 (8.84; 19.20)	0.262
Δ GH peak [µg/L]	7.54 (1.25; 11.29)	7.22 (0.90; 10.70)	7.82 (1.81; 11.95)	0.328
IGF-1 SDS at diagnosis	−1.23 (−1.94; −0.18)	−1.26 (−2.42; −0.33)	−1.12 (−1.72; −0.05)	0.135
IGF-1 SDS at retesting	−0.18 (−0.81; 0.58)	−0.23 (−0.84; 0.69)	−0.13 (−0.82; 0.51)	0.879
Δ IGF-1 SDS	0.99 (0.04; 2.11)	1.30 (0.17; 2.65)	1.04 (0.17; 1.91)	0.209
hSDS-0	−2.76 (−3.20; −2.43)	−2.81 (−3.26; −2.47)	−2.71 (−3.17; −2.27)	0.166
TH SDS	−1.08 (−1.58; −0.50)	−0.92 (−1.41; −0.46)	−1.21 (−1.70; −0.58)	**0.029**
corr hSDS-0	−1.70 (−2.31; −1.12)	−1.89 (−2.42; −1.43)	−1.58 (−2.19; −0.99)	**0.010**
hSDS-R	−1.02 (−1.54; −0.52)	−0.86 (−1.33; −0.33)	−1.26 (−1.70; −0.64)	**<0.001**
corr hSDS-R	−0.01 (−0.59; 0.57)	0.09 (−0.52; 0.62)	−0.01 (−0.60; 0.57)	**0.024**
Δ hSDS	1.74 (1.29; 2.28)	2.01 (1.56; 2.56)	1.58 (1.01; 2.06)	**<0.001**

* Except for the number of patients, values are expressed as the median and interquartile range (25th centile; 75th centile); *p*-values reported in the right column, concerning the differences between Groups PRE and POST, were calculated using the Mann–Whitney U test, significant differences are marked by bold fonts.

**Table 2 ijms-25-05739-t002:** The number of patients with normal and subnormal GH peaks at retesting according to cut-off values for persistent severe GHD, for transition period and for childhood, categorized into particular groups with respect to the moment of therapy withdrawal.

GH Peak [µg/L]	All (N = 260)	Group PRE (N = 121)	Group POST (N = 139)
<3.0	10 (3.85%)	9 (7.44%)	1 (0.72%)
3.0–6.0	24 (9.23%)	12 (9.91%)	12 (8.63%)
6.0–10.0	58 (22.31%)	97 (19.83%)	34 (24.46%)
>10.0	168 (64.61%)	76 (62.81%)	92 (66.19%)

**Table 3 ijms-25-05739-t003:** The number of boys and girls with normal and subnormal GH peaks at retesting according to cut-off values for persistent severe GHD, for the transition period and for childhood.

GH Peak [µg/L]	Boys (N = 173)	Girls (N = 87)
<3.0	7 (4.05%)	3 (3.45%)
3.0–6.0	13 (7.51%)	11 (12.65%)
6.0–10.0	29 (16.76%)	29 (33.33%)
>10.0	124 (71.68% *)	92 (66.19% *)

* significant (*p* = 0.008) difference between boys and girls.

**Table 4 ijms-25-05739-t004:** Pre-treatment characteristics, indices of rhGH therapy effectiveness and hormonal status at retesting in boys and girls.

	Boys	Girls	*p*
N (%) *	173 (66.5%)	87 (33.5%)	
Age at therapy onset [years]	13.8 (12.6; 14.8)	11.9 (10.7; 12.8)	**<0.001**
Age at retesting [years]	18.0 (17.5; 18.7)	16.1 (15.4; 16.9)	**<0.001**
Therapy duration [years]	4.3 (3.4; 5.3)	4.1 (3.3; 5.1)	0.442
GH peak at diagnosis [µg/L]	6.53 (5.28; 8.20)	7.20 (5.97; 8.28)	0.117
GH peak at retesting [µg/L]	14.20 (8.91; 19.30)	10.20 (7.19; 14.70)	**<0.001**
Δ GH peak [µg/L]	7.85 (2.60; 12.20)	2.96 (0.28; 7.50)	**<0.001**
IGF-1 SDS at diagnosis	−1.26 (-2.06; −0.24)	−1.05 (−1.80; −0.16)	0.314
IGF-1 SDS at retesting	−0.05 (−0.84; 0.61)	−0.34 (−0.79; 0.42)	0.261
Δ IGF-1 SDS	1.23 (0.11; 2.32)	0.98 (−0.24; 1.93)	0.267
hSDS-0	−2.84 (−3.30; −2.49)	−2.65 (−3.12; −2.24)	0.519
TH SDS	−1.00 (−1.58; −0.50)	−1.08 (−1.67; −0.50)	0.910
corr hSDS-0	−1.79 (−2.31; −1.18)	−1.62 (−2.32; −0.97)	0.330
hSDS-R	−1.07 (−1.45; −0.46)	−0.98 (−1.66; −0.58)	0.519
corr hSDS-R	0.03 (−0.59; 0.62)	0.00 (−0.58; 0.47)	0.640
Δ hSDS	1.81 (1.40; 2.31)	1.61 (1.15; 2.09)	**0.020**

* Except for the number of patients, values are expressed as the median and interquartile range (25th centile; 75th centile); *p*-values reported in the right column, concerning the differences between boys and girls, were calculated using the Mann–Whitney U test, significant differences are marked by bold fonts.

**Table 5 ijms-25-05739-t005:** The number of patients with normal and subnormal GH peaks at retesting according to cut-off values for severe GHD in adulthood, for transition period and for childhood categorized into particular groups according to the recent criteria for GHRS [15].

	Childhood	“Unlikely” GHD (N = 55)		“Doubtful” GHD (N = 96)		“True” GHD (N = 109)	
		Boys	Girls	Boys	Girls	Boys	Girls
Retesting		(N = 37)	(N = 18)	(N = 62)	(N = 34)	(N = 74)	(N = 35)
transGHD (N = 139)		34	15	58	28	64	30
childGHD (N = 15)		1	2	3	4	5	0
persistGHD (N = 16)		2	1	1	2	5	5

**Table 6 ijms-25-05739-t006:** Pre-treatment characteristics, indices of rhGH therapy effectiveness and hormonal status at retesting of patients with respect to the likelihood of GHD in childhood according to the current criteria [15].

	“Unlikely” GHD	“Doubtful” GHD	“True” GHD	*p*
N (boys/girls) *	55 (37/18)	96 (62/32)	109 (74/35)	
Age at therapy onset [years]	12.8 (11.3; 14.7)	13.2 (11.9; 14.2)	13.3 (12.0; 14.5)	**0.001**
Age at retesting [years]	17.3 (16.3; 18.5)	17.5 (16.3; 18.2)	17.7 (17.1; 18.4)	0.985
Therapy duration [years]	4.1 (2.9; 5.7)	4.1 (3.3; 5.1)	4.2 (3.6; 5.1)	**<0.001**
GH peak at diagnosis [µg/L]	6.41 (5.60; 7.70)	8.52 (7.68; 9.25)	5.60 (4.40; 6.28)	**<0.001**
GH peak at retesting [µg/L]	12.90 (9.30; 22.30)	12.60 (8.78; 19.05)	11.30 (7.17; 16.40)	**0.017**
Δ GH peak [µg/L]	7.70 (3.50; 16.00)	4.20 (0.69; 10.38)	6.22 (1.11; 11.19)	**0.018**
IGF-1 SDS at diagnosis	0.54 (0.37; 0.85)	−1.60 (−2.43; −1.20)	−1.33 (−2.09; −0.80)	**<0.001**
IGF-1 SDS at retesting	−0.41 (−0.93; 0.69)	−0.19 (−0.85; 0.57)	−0.09 (−0.75; 0.56)	0.991
Δ IGF-1 SDS	−1.12 (−1.56; 0.05)	1.60 (0.61; 2.78)	1.50 (0.60; 2.22)	**<0.001**
hSDS-0	−2.88 (−3.47; −2.38)	−2.76 (−3.18; −2.47)	−2.70 (−3.17; −2.28)	0.272
TH SDS	−1.39 (−1.68; −0.75)	−0.92 (−1.51; −0.49)	−1.06 (−1.58; −0.50)	0.109
corr hSDS-0	−1.57 (−2.28; −0.99)	−1.86 (−2.39; −1.17)	−1.66 (−2.28; −1.03)	0.468
hSDS-R	−1.45 (−2.06; −0.96)	−0.91 (−1.45; −0.40)	−0.96 (−1.47; −0.46)	**<0.001**
corr hSDS-R	−0.15 (−0.73; 0.12)	0.12 (−0.62; 0.78)	0.08 (−0.52; 0.61)	**0.020**
Δ hSDS	1.41 (0.91; 2.08)	1.94 (1.48; 2.30)	1.69 (1.34; 2.31)	**0.010**

* Except for the number of patients, values are expressed as the median and interquartile range (25th centile; 75th centile); *p*-values reported in the right column, concerning the differences between the groups with “True”, “Doubtful” and “unlikely” GHD, were calculated using the Kruskall–Wallis ANOVA test., significant differences are marked by bold fonts.

**Table 7 ijms-25-05739-t007:** The number of patients with persistent or non-transient and transient GHD peaks at retesting in the groups of patients divided according to the criteria, including the pre-test likelihood of GHD in childhood, proposed by Wit et al. [16].

	Childhood	“Pre-Test Unlikely” GHD (N = 168)		“Excluded” GHD (N = 45)		“Confirmed” GHD (N = 47)	
		Boys	Girls	Boys	Girls	Boys	Girls
Retesting		(N = 110)	(N = 58)	(N = 62)	(N = 32)	(N = 74)	(N = 35)
T-GHD (N = 247)		107	57	32	10	25	16
NT-GHD (N = 3)		1	1	1	0	0	0
P-GHD (N = 10)		0	2	0	2	5	1

**Table 8 ijms-25-05739-t008:** Pre-treatment characteristics, indices of GH therapy effectiveness and hormonal status at retesting of patients with respect to the diagnosis, including the pre-test likelihood of GHD in childhood, according to the criteria proposed by Wit et al. [16].

	“Pre-Test Unlikely” GHD	“Excluded” GHD	“Confirmed” GHD	*p*
N (boys/girls) *	168 (110/58)	45 (33/12)	47 (30/17)	
Age at therapy onset [years]	12.8 (11.6; 14.2)	13.7 (12.7; 14.4)	13.7 (12.1; 14.8)	0.471
Age at retesting [years]	17.4 (16.4; 18.4)	17.8 (17.0; 18.2)	17.7 (17.0; 18.3)	0.233
Therapy duration [years]	4.3 (3.4; 5.4)	4.1 (3.4; 4.6)	4.0 (3.4; 5.1)	0.425
GH peak at diagnosis [µg/L]	6.75 (5.63; 8.09)	8.69 (7.60; 9.38)	4.84 (3.73; 5.97)	**<0.001**
GH peak at retesting [µg/L]	13.90 (9.06; 20.15)	12.40 (8.37; 17.80)	8.40 (4.06; 13.00)	**0.017**
Δ GH peak [µg/L]	7.38 (2.33; 13.18)	3.72 (-0.68; 8.57)	4.03 (0.62; 8.10)	**0.018**
IGF-1 SDS for CA at diagnosis	−0.59 (−1.24; 0.35)	−2.45 (−3.03; −1.91)	−2.48 (−3.30; −1.64)	**<0.001**
IGF-1 SDS for BA at diagnosis	0.14 (−0.48; 0.78)	−1.73 (−2.42; 1.27)	−1.52 (−2.35; 1.23)	**<0.001**
IGF-1 SDS for CA at retesting	0.04 (−0.58; 0.69)	−0.46 (−1.04; 0.33)	−0.46 (−0.95; 0.56)	0.069
Δ IGF-1 SDS for CA	0.52 (−0.69; 1.55)	2.15 (1.34; 3.29)	2.06 (0.87; 3.07)	**<0.001**
hSDS-0	−2.71 (−3.17; −2.38)	−2.81 (−3.32; −2.53)	−2.84 (−3.24; −2.28)	0.425
TH SDS	−1.08 (−1.67; −0.50)	−1.08 (−1.57; −0.58)	−1.07 (−1.58; −0.33)	0.685
corr hSDS-0	−1.65 (−2.31; −1.13)	−1.76 (−2.27; −1.14)	−1.78 (−2.35; −0.97)	0.964
hSDS-R	−1.18 (−1.66; −1.04)	−0.79 (−1.34; −0.21)	−0.76 (−1.42; −0.29)	**0.011**
corr hSDS-R	−0.04 (−0.63; 0.36)	0.37 (−0.22; 0.88)	0.12 (−0.59; 0.70)	**0.009**
Δ hSDS	1.58 (1.13; 2.14)	2.12 (1.73; 2.51)	2.02 (1.46; 2.66)	**<0.001**

* Except for the number of patients, values are expressed as the median and interquartile range (25th centile; 75th centile); *p*-values reported in the right column, concerning the differences between the groups, were calculated using the Kruskall–Wallis ANOVA test, significant differences are marked by bold fonts.

## Data Availability

The raw data supporting the conclusions of this article will be made available by the authors on request, further inquiries can be directed to the corresponding author.

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
