# Peer review of "Transient Isolated, Idiopathic Growth Hormone Deficiency—A Self-Limiting Pediatric Disease with Male Predominance or a Diagnosis Based on Uncertain Criteria? Lesson from 20 Years’ Real-World Experience with Retesting at One Center"

_ijms, 2024, doi:10.3390/ijms25115739_

Round 1

Reviewer 1 Report

Comments and Suggestions for Authors

The manuscript “Transient isolated idiopathic growth hormone deficiency – a self-limiting pediatric disease with male predominance or a diagnosis based on uncertain criteria? Lesson from 20-years’ real-world experience with retesting in one center” by Smyczynska et al, presents a timely and valuable discussion in determining the decision of hGH therapy withdrawal in the appropriate patient population. The authors address a critical issue and provide several valuable recommendations for appropriate patient selection for rhGH treatment. The sex-specific analyses are particularly valuable. I have nothing to add or take away from this work. There are minor grammatical errors as in line 45 (withdrawal of..), line-81 (falsely positive?). Please check throughout.

Comments on the Quality of English Language

the English language in the manuscript is of acceptable for publication, following minor typos and grammatical editing.

Author Response

Dear Reviewer,

thank you for the positive review of our paper. Gramatical errors are corrected as suggested (green highlights in text; yellow highlights are related to the comments of Reveiwer 2). The paper has been checked for any gramatical or spelling errors and corrected.

Kind regards,

Authors

Reviewer 2 Report

Comments and Suggestions for Authors

Dear Authors,

I have reviewed your paper titled "Transient isolated idiopathic growth hormone deficiency – a self-limiting pediatric disease with male predominance or a diagnosis based on uncertain criteria? Lesson from 20-years’ real-world experience with retesting in one center" and would like to provide feedback on several points regarding the content:

  1. It would be beneficial to consider moving the content from lines 130-132 of the Results section to the Methods section for better clarity and organization.
  2. In Table 1, please ensure correct usage of "-" and ";" symbols. There are typos present. Additionally, providing a comprehensive legend for the entire figure, distinguishing parts with labels such as a, b, c, would enhance clarity.
  3. For consistency, please ensure uniform text formatting (such as bold) across all tables or consider removing bold formatting altogether.
  4. There appears to be an error in the figure labeling within Figure 2. Kindly review and make the necessary corrections.
  5. Could you please clarify the meaning of "XX" within the parentheses in line 510 of the Materials and Methods section?
  6. There is redundancy in the punctuation in line 626 of the Conclusions section. Additionally, I recommend revising the conclusion to accurately reflect the research design and results.
  7. Lastly, there is a typographical error in the Funding section. Please review and make the necessary corrections.
Comments on the Quality of English Language

Overall, I recommend conducting a thorough proofreading of the entire manuscript to identify and correct any English errors or typos. English language editing may also be necessary to ensure clarity and precision in the manuscript.

Thank you for your attention to these points. I look forward to seeing the revisions.

Author Response

Dear Reveiwer,

thank you for the review and very detailed comments. All your recommendations have been taken into account (suggested modifications of the text are highlighted in yellow)

  1. Text in lines 130-132 has been just removed as a sentence with the same information is repreated at the beginning of section "Methods".
  2. Errors in using "-" and ":" have been corrected (thank you once more for thorough correction).
  3. According to your suggestion, formatting of the Tables has been unified for consistency by removing bold formatting.
  4. An error in labeeling of Figure 2 has been corrcted.
  5. This is a standard recording of a normal female karyotype with 46 chromosomes in total, including two X chromosomes. We would like to report this for clear confirmation that all girls included had no abnormalities detectable in karyotype examination.
  6. Thank you, unnecessary punctuation has been removed. Conslusions have been partially modified in order to more to accurately reflect only the research design and results (lines 622-624).
  7. "Funding" section has been corrected by removing unnecessary text.

The manuscript has been cerfully read once more and some corretions or clarifications of the sentences have been made (see yellow highlights - lines 27, 57, 59, 60, 61, 93, 102, 117, 141, 239, 286, 296, 299, 301, 374, 376, 4-5-406, 417, 471, 546). Introducing abbreviations was placed at their first use for IGF-1, hSDS-R, hSDS-0 and MRI  (lines 61, 141, 146, 385).

We would like to assure you that we have made every effort to make all necessary corrections in the manuscript.

Kind regards,

Authors

Round 2

Reviewer 2 Report

Comments and Suggestions for Authors

The requested changes have been successfully modified to reflect the desired content. Thank you for your hard work.